# Unveiling Replication Timing-Dependent Mutational Biases: Mechanistic Insights from Gene Knockouts and Genotoxins Exposures

**DOI:** 10.3390/ijms26157307

**Published:** 2025-07-29

**Authors:** Hadas Gross-Samuels, Amnon Koren, Itamar Simon

**Affiliations:** 1Department of Microbiology and Molecular Genetics, IMRIC, Faculty of Medicine, Hebrew University of Jerusalem, Jerusalem 91120, Israel; hadas.samuels@mail.huji.ac.il; 2Department of Molecular and Cellular Biology, Roswell Park Comprehensive Cancer Center, Buffalo, NY 14263, USA

**Keywords:** replication timing, mutational signatures, DNA damage, DNA repair

## Abstract

Replication timing (RT), the temporal order of DNA replication during S phase, influences regional mutation rates, yet the mechanistic basis for RT-associated mutagenesis remains incompletely defined. To identify drivers of RT-dependent mutation biases, we analyzed whole-genome sequencing data from cells with disruptions in DNA replication/repair genes or exposed to mutagenic compounds. Mutation distributions between early- and late-replicating regions were compared using bootstrapping and statistical modeling. We identified 14 genes that exhibit differential effects in early- or late-replicating regions, encompassing multiple DNA repair pathways, including mismatch repair (*MLH1, MSH2, MSH6, PMS1*, and *PMS2*), trans-lesion DNA synthesis (*REV1*) and double-strand break repair (*DCLRE1A* and *PRKDC*), DNA polymerases (*POLB, POLE3,* and *POLE4*), and other genes central to genomic instability (*PARP1* and *TP53*). Similar analyses of mutagenic compounds revealed 19 compounds with differential effects on replication timing. These results establish replication timing as a critical modulator of mutagenesis, with distinct DNA repair pathways and exogenous agents exhibiting replication timing-specific effects on genomic instability. Our systematic bioinformatics approach identifies new DNA repair genes and mutagens that exhibit differential activity during the S phase. These findings pave the way for further investigation of factors that contribute to genome instability during cancer transformation.

## 1. Introduction

Somatic mutations in cancer genomes accumulate throughout cell lineage development, reflecting the combined effects of diverse mutational processes [1]. These processes generate distinct mutational signatures—characteristic combinations of mutation types that serve as fingerprints of specific mutagenic mechanisms [2,3]. Systematic analyses of mutation distributions across genomic contexts have revealed numerous signatures with varying biological origins [4]: some, like single base substitution 1 (SBS1), are ubiquitous across tumors, while others (e.g., SBS7a–d) are tumor type-specific and linked to environmental exposures like UV radiation. Recent advances extend beyond single base substitutions to include doublet base substitutions (DBS) and small insertions/deletions (INDELs), broadening our understanding of mutational dynamics [5].

DNA replication critically influences mutagenesis by converting DNA damage into permanent mutations during S phase [6]. Moreover, various features of the replication program seem to influence mutation load. Replication-fork rate and direction correlate with mutation loads [7,8,9]. Replication timing (RT)—the temporal order of DNA replication during S phase [10]—is a key determinant of mutation rates [9]. Early-replicating regions (ERR) typically exhibit open chromatin, high transcriptional activity, and lower mutation rates, while late-replicating regions (LRR) display compact chromatin and elevated mutation rates in both germline and somatic cells [11]. These differences highlight the interplay between replication dynamics and mutagenic mechanisms.

Our prior analysis of 2787 whole-genome sequenced tumors revealed signature-specific RT associations: signatures like SBS7b associate with ERR, whereas SBS7a aligns with LRR [12]. This raises a fundamental question: what mechanisms drive differential frequencies of specific mutation types between ERR and LRR? While enhanced repair efficiency in accessible ERR regions is a plausible explanation, it is rather general and does not explain the differences between different mutational processes. Moreover, recent work demonstrates that chromatin accessibility alone cannot fully explain these differences [7,8]. Notably, signatures like SBS5, SBS16, and SBS40 are enriched in ERR despite greater accessibility, and signatures such as SBS8 and SBS40 maintain RT associations even after accessibility normalization [12]. These findings implicate additional cellular processes beyond chromatin packaging in RT-dependent mutagenesis.

Previous analyses of cells harboring defects in mismatch repair (MMR) or global genome nucleotide excision repair (GG-NER) [13,14] showed that the elevated mutation rates in LRR depend on these pathways. We extended this approach by taking advantage of isogenic systems, in which the effect of defined perturbations on mutagenicity was determined by whole-genome sequencing (WGS). We hypothesized that disruptions in specific repair pathways would selectively alter ERR/LRR mutation biases, revealing their RT-dependent roles. By integrating data from knockout models [15] and genotoxin exposures [16] with replication timing profiles [17], we bioinformatically identified DNA repair pathways whose dysfunction disrupts RT-mutation relationships, providing mechanistic insights into replication timing-specific mutagenesis and its implications for genome instability in cancer [18,19,20].

## 2. Results

### 2.1. Development of a Methodology to Assess Gene Knockout Effects on Mutational Spectra and Burden

To investigate the contribution of DNA replication and repair genes to replication timing (RT)-dependent mutagenesis, we utilized data from a recent study [15] profiling mutations in hiPSCs with knockouts (KOs) in 42 DNA replication and repair genes (Figure 1a). We developed a comprehensive analytical framework to identify genes whose disruption leads to significant changes in mutational spectra and burden, following methods described in Zou et al. [21], allowing further stratification by RT.

For mutational spectrum analysis, we generated single base substitution (SBS) profiles using SigProfilerMatrixGenerator [22], capturing all 96 substitution types within their trinucleotide contexts (Figure 1b). Each KO’s mutation spectrum was compared to that of control cells. To address the challenge of limited sample sizes—particularly the small number of control samples—we implemented a bootstrap resampling strategy. We applied this strategy exclusively to control profiles, unlike previous methods that resampled both control and KO samples [15,16,21]. Our rationale was to maintain the full statistical power of the KO data while generating a robust and stable estimate of the natural variability in the control spectra. The resulting bootstrapped distribution of control spectra enabled precise quantification of spectrum divergence using multiple testing-corrected *p*-values (significance threshold ≤ 0.05). Our results were consistent with published findings [15], identifying all mismatch repair (MMR) genes (*MLH1*, *MSH2*, *MSH6*, *PMS1*, and *PMS2*) as having significantly altered mutational spectra, as well as *REV1*, a member of the trans-lesion synthesis DNA polymerase Y family (Figure 1c).

For mutation count analysis, we applied the Mann-Whitney U test to directly compare KO mutation burdens against control baselines (Figure 1d). To improve statistical power given limited control samples, we expanded the control group by including KO replicates that showed no significant difference from the original controls in initial comparisons. This analysis revealed that several genes with altered mutational spectra also exhibited significantly different mutation burdens: MMR genes (*MLH1*, *MSH2*, *MSH6*, and *PMS2*) showed elevated burden, while *PMS1* displayed spectrum changes without burden alterations. *REV1* exhibited both an altered spectrum and reduced mutation burden compared to controls (Figure 1e). Taken together, our analytical approach yielded results highly similar to those reported in the original study [15].

### 2.2. Identification of Genes Affecting Mutation Rate and Spectra in a Replication Timing-Dependent Manner

To assess the differential contribution of DNA repair genes to genome stability in early- and late-replicating regions, we partitioned each dataset into early-replicating regions (ERR) and late-replicating regions (LRR), using hiPSC replication timing profiles (Figure 2a). We then applied our established analytical pipeline for SBS analysis separately to each RT state, generating and normalizing mutational profiles by trinucleotide content and performing both spectrum and mutation count analyses (Figure 2b–d).

This approach identified KO samples that showed significant deviation from controls in either ERR or LRR (Figure 2b–e and Table 1). We found that the mismatch repair (MMR) genes (*MLH1*, *MSH2*, *MSH6*, *PMS1*, and *PMS2*) affected the mutational spectrum only in LRR, suggesting increased activity of these pathways in late-replicating regions, consistent with previous reports [13]. In addition, genes involved in trans-lesion synthesis (*REV1*), double-strand break repair (*DCLRE1A*, *PRKDC*), as well as *POLE3* [23], which is involved in chromatin integrity during DNA replication, and the tumor suppressor gene, *TP53*, also specifically influenced the LRR. Interestingly, only the KO in a gene involved in trans-lesion synthesis (*REV1*) showed effect on mutational burden.

We next analyzed the effects of KOs on INDELs. Due to the low frequency of INDELs in the dataset, spectrum profiling was not feasible; however, we were able to assess INDEL burden. This analysis revealed that *POLE4* and *PMS1* exhibited altered mutation burden exclusively in LRR, whereas *POLB* and, to a lesser extent, *PARP1* showed increased mutation burden only in ERR (Figure 2e and Table 1).

### 2.3. Identification of External Treatments Affecting Mutation Burden and Spectra in a Replication Timing-Dependent Manner

After analyzing the RT-dependent effects of DNA replication and repair gene knockouts, we next examined the impact of DNA-damaging agents. We utilized a dataset profiling mutations in hiPSCs exposed to 77 chemical carcinogens, therapeutic agents, or DNA damage response (DDR) inhibitors and two sources of radiation, damaging DNA in various ways [16] (Figure 3a). Applying the same analytical pipeline as for the knockout data, we assessed both mutational spectra and burden for each treatment. Consistent with published findings [16], our analysis revealed that 55 out of 79 conditions exhibited significant changes in either mutation burden or spectrum (Appendix A).

Stratification by replication timing uncovered subtle RT-specific effects. We have seen mostly effects on mutation burden, and for most compounds, it was RT-independent (Figure 3b). Few treatments showed RT-dependent effects on mutation burden. Aristolochic acid II (AAII; 3.75 µM), a compound known to form adducts with purines [24], and styrene oxide (75 µM), a compound causing a variety of transition and transversion mutations [25], displayed elevated mutation burden preferentially in early-replicating regions (ERR). In contrast, methyleugenol (1.25 µM) and N-nitrosopyrrolidine (50 mM), a constituent of tobacco smoke [26], showed increased mutation burden preferentially in late-replicating regions (LRR). Olaparib (0.625 µM), a poly ADP ribose polymerase (PARP) inhibitor [27], was associated with reduced mutation burden exclusively in LRR (Figure 3b and Table 2). For few additional treatments (Figure 3b and Table 2), the differential activity in ERR versus LRR was only marginal. While they passed our stringent threshold (adjusted *p*-value < 0.05) in one RT state, they were close to significant also in the other state. INDEL analysis, though limited by data sparsity, identified dibenzo[a,l]pyrene (DBP, 0.0313 µM), a constituent of tobacco smoke [28], as the only compound with RT-specific effects, increasing INDEL counts in ERR (Figure 3b and Table 2). These findings demonstrate that exogenous mutagens can modulate mutation burden—and to a lesser extent, mutation spectrum—in a replication timing-dependent manner.

### 2.4. Power Analysis

To gain additional confidence in our findings, we performed simulation-based analyses (see Section 4) to estimate the statistical power of each conclusion. This approach confirmed that both knockout lines and compounds showing significant changes in mutation burden (*p* ≤ 0.05) are indeed robust and reliable. However, some of the cases we initially classified as non-significant may reflect limited statistical power, as they reached significance in a substantial fraction (>50%) of the simulations (Appendix A).

## 3. Discussion

In this study, we systematically investigated the impact of DNA replication timing on mutational processes by analyzing whole-genome sequencing data from 42 DNA replication and repair gene knockouts [15] and 79 environmental mutagen exposures [16] in human-induced pluripotent stem cells. Our approach advances previous work in several important ways. First, we performed a comprehensive, side-by-side comparison of a large panel of genetic and environmental perturbations, enabling direct assessment of RT-dependent effects on both mutational spectra and burden. Second, we leveraged a high-resolution RT map generated from the same cell type [29], which is critical for accurately correlating RT with mutation distributions, as highlighted in recent studies demonstrating the importance of cell type-matched RT data for interpreting mutational landscapes. Third, our analytical framework systematically evaluated both mutational spectrum and mutation count changes across early- and late-replicating regions, providing a nuanced view of how different pathways and exposures shape genome instability.

Through this systematic analysis, we identified 14 genes and 19 compounds that exhibit differential effects in early- or late-replicating regions. Notably, our findings reinforce and extend previous observations regarding the mismatch repair (MMR) pathway. Consistent with prior reports, we found that disruption of core MMR genes (*MLH1*, *MSH2*, *MSH6*, *PMS1*, and *PMS2*) predominantly affects the mutational spectrum and burden in late-replicating regions (LRR), supporting the model that MMR activity is heightened or more essential in these genomic compartments. This is in line with earlier studies demonstrating increased mutation rates in LRR upon MMR deficiency [13] and with the broader understanding that DNA repair pathway activity can vary across the replication timing landscape.

Our analysis also highlighted the RT-dependent roles of other DNA repair and damage tolerance pathways. REV1 knockout resulted in a distinct mutational spectrum and a significant reduction in mutation burden specifically in LRR, consistent with its function as a trans-lesion synthesis polymerase active during G2/M phases [30,31]. *DCLRE1A* (*SNM1A*) knockout was associated with reduced mutation counts in LRR, potentially linked to its peak protein levels during mitosis [32], though further experimental validation is needed to confirm these mechanistic links.

Importantly, several of our findings are novel and warrant further investigation. For example, the identification of *POLE3* and *TP53* as genes with RT-dependent mutagenic effects, and the observation that certain INDEL-related genes (such as *POLE4*, *POLB*, and *PARP1*) display region-specific mutation burdens, suggest previously unappreciated complexity in how DNA repair and replication factors interact with the replication timing program. Experimental validation of these findings, including mechanistic studies of protein expression, localization, and activity across the cell cycle, will be essential to fully elucidate their roles.

Our bioinformatics analyses have identified several compelling candidate factors potentially involved in replication timing-dependent mutagenesis. While these findings provide a valuable framework, elucidating the exact molecular mechanisms will require further experimental validation. In particular, investigating the temporal dynamics of candidate protein levels and their chromatin distribution during S phase will be essential. We are planning to pursue such experiments in future studies to build on the insights presented here. An important question is whether the replication timing-dependent mutational patterns we identified can also be observed in primary tumors. However, the complexity and heterogeneity of tumor genomes—characterized by multiple co-occurring mutations and diverse mutational processes—pose a significant challenge in isolating the specific signatures that are readily detectable in controlled, isogenic systems. This limitation highlights the value of using defined perturbation models to uncover mechanistic insights into mutagenesis and underscores the need for future strategies to deconvolve these effects in clinical samples.

Our study also integrated the analysis of environmental mutagens, revealing that exogenous agents can modulate mutation burden—and to a lesser extent, mutational spectrum—in a RT-dependent manner. While most compounds exhibited genome-wide effects, a subset demonstrated distinct biases toward early-replicating regions (ERR) or late-replicating regions (LRR), raising important mechanistic questions.

These differential effects may reflect several overlapping biological processes. One possibility is that the distribution of DNA damage itself is non-uniform across the genome. For example, polycyclic aromatic hydrocarbons such as benzopyrene have been shown to preferentially form DNA adducts in open chromatin regions, which are often associated with ERR [33]. On the contrary, UV damage is evenly distributed along the genome, and the accumulation of UV-lesions in LRR can be explained by efficient repair in ERR due to increased accessibility of the nucleotide excision repair machinery [34]. These general principles may explain our findings: we observed that compounds such as AAII, styrene oxide, and dibutyl phthalate (DBP) induced a disproportionately high mutation burden in ERR, suggesting preferential damage formation in the accessible ERR of the genome. In contrast, methyleugenol and N-nitrosopyrrolidine (NPYR) showed higher mutation loads in LRR, which may reflect more persistent DNA lesions in heterochromatic regions, where repair is generally less efficient. The absence of strong ERR mutagenesis from these agents also suggests that ERR may have been more effectively repaired under these conditions, pointing to the importance of differential repair capacity across the RT axis.

These findings highlight the complex interplay between chromatin state, DNA damage formation, and repair efficiency. Chromatin accessibility in ERR likely facilitates both damage and repair, whereas the compact structure of LRR may limit both damage exposure and repair access, potentially leading to lesion persistence and mutagenesis over time. This layered interaction between damage and repair may determine whether a compound leads to elevated mutations in ERR or LRR. Further experiments are needed to decipher the exact mechanism in each case.

Interestingly, olaparib—a PARP inhibitor rather than a direct damaging agent—caused a selective reduction in mutation counts in LRR. This stands in contrast to PARP1 knockout, which slightly elevated INDELs in ERR. While both perturbations ultimately increased the relative mutational burden in ERR, their differing effects suggest distinct contributions of PARP activity across replication timing domains. These differences may reflect varying roles of PARP1 and PARP2 in responding to replication-associated DNA lesions in different chromatin contexts. Future studies combining genetic and pharmacologic perturbations of PARP function will be critical to disentangling these roles and clarifying how DNA repair dependencies vary across the RT landscape.

Together, these observations emphasize that the RT-dependent mutagenic effects of environmental agents are shaped by a combination of damage distribution, chromatin context, repair efficiency, and possibly lesion persistence—underscoring the importance of studying mutagenesis in the framework of genome organization and replication timing.

Genomic instability is one of the cancer hallmarks [35], and better understanding of the factors that affect mutation distribution is essential for deciphering the processes that lead to cancer transformation.

In summary, our work demonstrates that replication timing is a critical modulator of mutagenesis, with distinct DNA repair pathways and environmental agents exhibiting region and RT-specific effects on genomic instability. By systematically comparing genetic and environmental perturbations using high-resolution, cell type-matched RT maps, we provide a framework for dissecting RT-associated mutational processes. These findings have broad implications for understanding genome evolution, cancer development, and the design of targeted interventions to preserve genome integrity.

## 4. Materials and Methods

### 4.1. Data Sources

We analyzed somatic mutation data from published studies employing CRISPR–Cas9 knockouts and mutagen exposures in human cell lines. Specifically, we used whole-genome sequencing (WGS) data from human-induced pluripotent stem cells (hiPSCs) with CRISPR–Cas9 knockouts of 42 DNA replication and repair genes, as well as an unrelated control gene (*ATP2B4*) [15]. For each gene knockout, subclones were derived, grown for multiple-generation WGS, and somatic mutations were called after subtracting parental clone variation [15]. We also used WGS data from hiPSCs exposed to 79 environmental carcinogens and controls, with subclones sequenced and analyzed as above [16].

Cell-specific RT data for hiPSCs were obtained from Ding et al. [17].

### 4.2. Mutational Profile Generation

Mutational profiles were generated using SigProfilerMatrixGenerator v1.2.15 [22]. Mutation call files were converted to Mutation Annotation Format (MAF) using custom Python scripts (see Section 4.8), ensuring compatibility with SigProfiler. Single base substitution (SBS) profiles were constructed using the standard 96-channel trinucleotide convention. INDEL profiles were generated using the standard 28-channel categories.

### 4.3. Replication Timing Stratification

For each dataset, mutations were stratified according to replication timing. The median RT value was used to distinguish early- and late-replicating regions. Each MAF file was partitioned accordingly, resulting in RT-specific mutational profiles for early and late RT regions. These mutational profiles served as the foundational input for subsequent statistical analyses and bootstrapping comparisons to identify significant deviations between experimental and control profiles in different RT regions of the genome.

### 4.4. Trinucleotide Normalization

To account for differences in trinucleotide composition across RT regions, we normalized the mutation frequencies by the trinucleotide frequencies, separately for ERR and LRR.

### 4.5. Bootstrapping Method for Qualitative Identification of Significant Differences in Mutational Spectra

To assess whether knockout (KO) mutational spectra differed significantly from controls, we applied a bootstrapping approach following Zou et al. [21]. We calculated the centroid of normalized control samples and constructed a mutation pool where each mutation type’s frequency corresponded to its count across control replicates. We generated 10,000 bootstrapped control replicates by randomly sampling mutations from this pool, ensuring that each control sample contributes equally to the total replicates. The number of mutations drawn per replicate matched the original control sample’s mutation count.

For each normalized KO sample, we calculated its Euclidean distance from the control centroid. Significance was determined by comparing this distance to the distribution of distances from bootstrapped control replicates, with *p*-values defined as the proportion of bootstrapped samples with greater distances. *p*-values were corrected for multiple comparisons using the Benjamini-Hochberg method. KO samples with corrected *p*-values ≤ 0.05 were considered significantly different from controls.

### 4.6. Statistical Method for Quantitative Identification of Significant Differences in Mutation Counts

To identify KO samples with significantly altered mutation counts, we used the Mann-Whitney test in a two-stage approach. First, we compared mutation counts from each KO replicates against all control replicates using a two-sided Mann-Whitney test, performed separately for each RT state (all, early, and late).

To construct a more robust control set, we expanded the original controls by adding KO samples showing no significant mutation count changes (*p* > 0.1333 in the ALL dataset).

Using these expanded control sets, we repeated the Mann-Whitney test for each RT condition and applied Benjamini-Hochberg correction for multiple comparisons. KO samples with adjusted *p*-values ≤ 0.05 were considered to have significantly altered mutation counts.

### 4.7. Power Analysis

To estimate the statistical power of each conclusion, we conducted simulation-based analyses. For each KO strain and compound, we generated 20 simulated replicates by randomly selecting samples with mutation burdens within the range observed in the original dataset. Each replicate was processed through our analysis pipeline to evaluate changes in mutation burden. The statistical power was estimated as the fraction of replicates that exhibited a significant deviation from control samples (adjusted *p* < 0.05) in either the ERR or LRR metric.

### 4.8. Software and Code Availability

Analyses were performed using Python 3.10, SigProfilerMatrixGenerator v1.2.15 [22], and custom scripts available at https://github.com/hadasamuels/RT_mutagenesis_article_code.git, accessed on 26 June 2025.

## Figures and Tables

**Figure 1 ijms-26-07307-f001:**
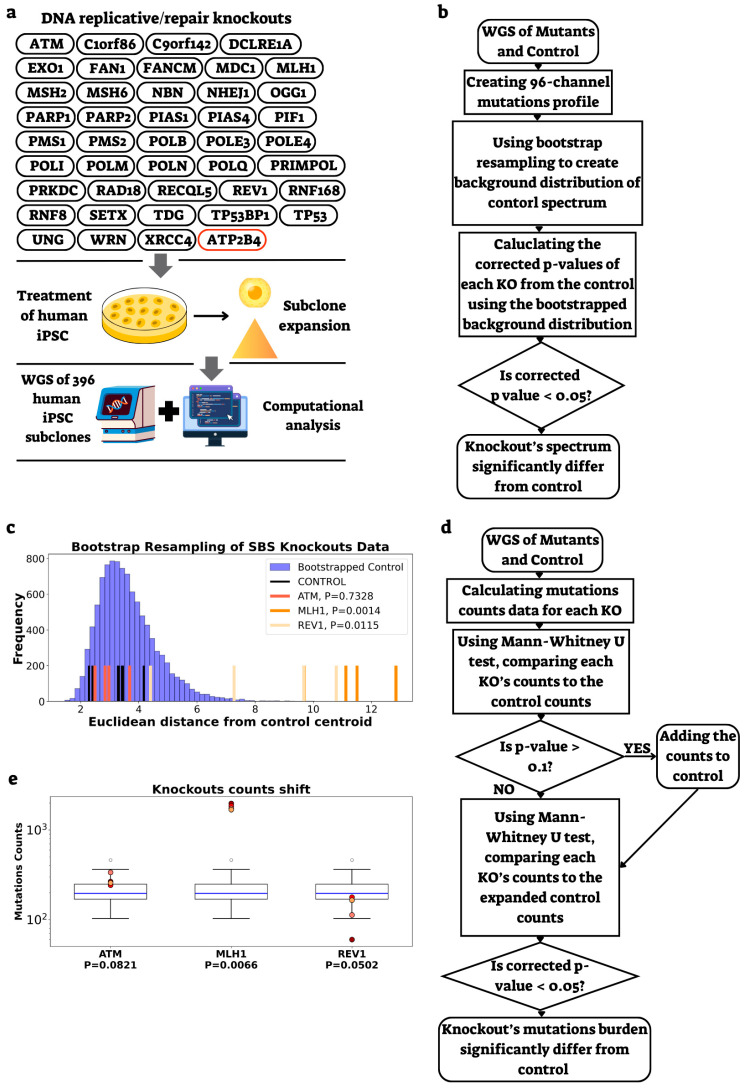
Methodology. (**a**) Schematic representation of the KO dataset, highlighting the control gene (*ATP2B4*, in red), and the experimental workflow from subclone generation to computational analysis. (**b**) Diagram outlining the process from WGS of KO and control samples, through mutational profile generation, to the identification of significant differences in mutational spectra using bootstrap resampling. (**c**) Histogram of Euclidean distances for bootstrapped control replicates (blue), original control samples (black), and three example KOs (*ATM, MLH1*, and *REV1*; red shades). Corrected *p*-values (see Section 4) indicate significance of spectrum changes. *ATM* KO shows a non-significant change, while *MLH1* and *REV1* KOs are significant. (**d**) Schematic illustrating the process from WGS data to detection of significant differences in total mutation counts between KOs and controls, including the use of an expanded control group. (**e**) Boxplots depicting mutation count distributions for the expanded control group, with individual KO replicates shown as colored dots. Corrected *p*-values (two-sided Mann-Whitney test) compare each KO to the expanded control. For detailed results, including INDEL analysis (see Appendix A).

**Figure 2 ijms-26-07307-f002:**
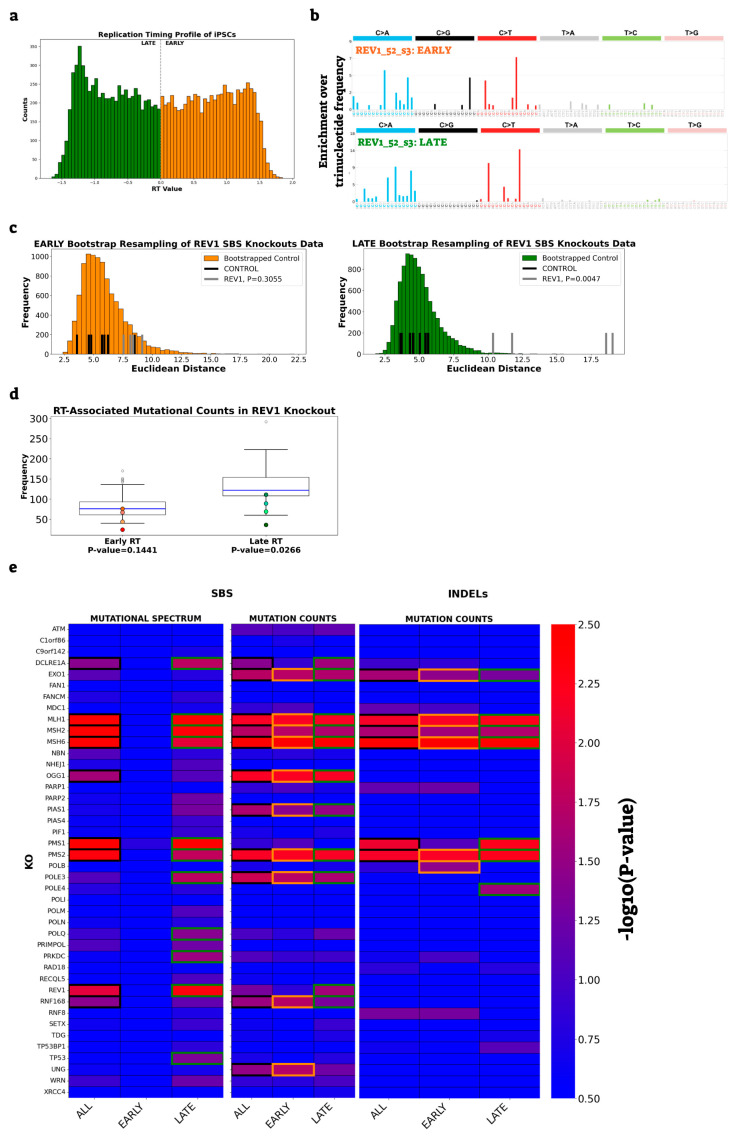
Replication timing-dependent mutational effects in iPSC knockouts. (**a**) Histogram showing the distribution of RT values in hiPSCs. The dashed grey line separates early- (orange) and late- (green) replicating regions. (**b**) Normalized 96-channel SBS profiles for the *REV1* knockout, shown separately for early- (top) and late- (bottom) replication timing regions. (**c**) Euclidean distances of bootstrapped control replicates (histograms), original controls (black lines), and *REV1* knockout replicates (gray lines), shown for early- (ERR, orange histogram, left) and late- (LRR, green histogram, right) replicating regions. The geometric mean of corrected *p*-values is indicated. (**d**) Boxplots display the distribution of mutation counts for the expanded control group in early and late RT regions. As expected, mutation burden is higher in LRR for the control samples. Mutation counts for *REV1* knockout replicates are overlaid as colored dots (orange for ERR, green for LRR), with corrected *p*-values from two-sided Mann–Whitney tests shown. Notably, *REV1* knockout reduces mutation burden specifically in LRR. (**e**) Heatmaps of −log10 (corrected *p*-values) for SBS mutational spectrum and mutation counts across all, early, and late RT regions (**left**) and for INDEL counts (**right**). Significant results are highlighted by colored edges (black: all RT; orange: ERR; green: LRR). Color intensity reflects the −log10 (corrected *p*-value). For detailed results, see Appendix A.

**Figure 3 ijms-26-07307-f003:**
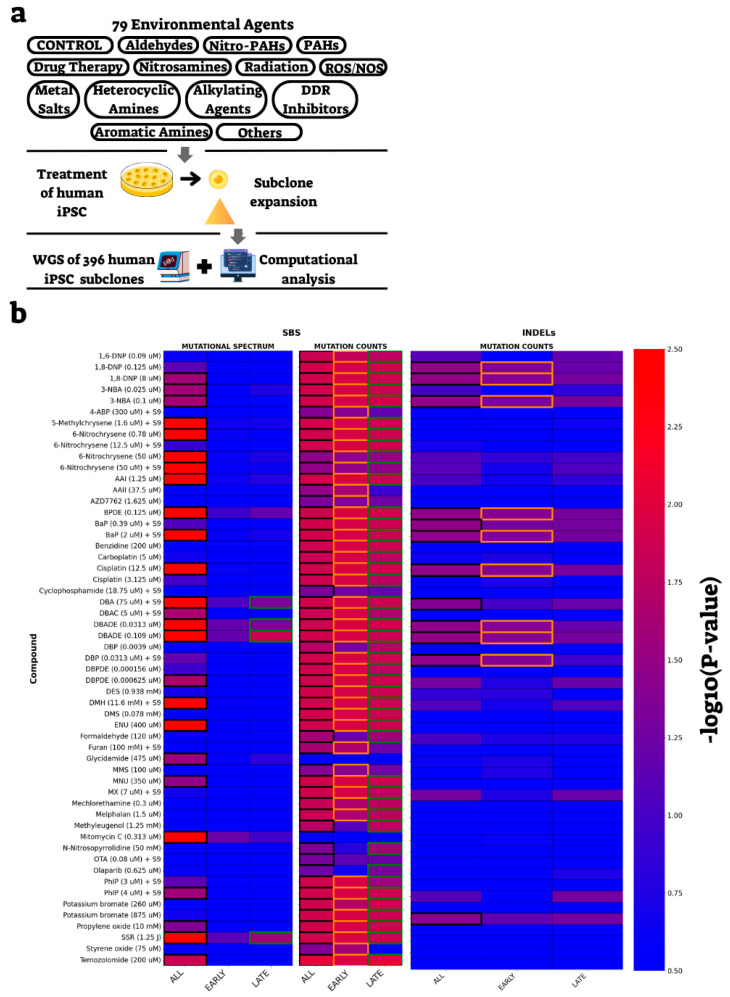
RT-dependent effects of environmental mutagens in human iPSCs. (**a**) Schematic representation of the experimental design for mutagenic compound analysis, including the major classes of environmental agents tested. The full list of compounds is provided in Appendix A. (**b**) Heatmaps displaying −log10 (corrected *p*-values) for SBS mutational spectrum and mutation counts across all, early, and late replication timing regions (left) and for INDEL mutation counts (right). Colored edges indicate statistical significance (black: all RT regions; orange: early-replicating regions [ERR]; green: late-replicating regions [LRR]). Color intensity reflects the −log10 (corrected *p*-value). For comprehensive results, see Appendix A.

**Table 1 ijms-26-07307-t001:** Genes exhibiting replication timing-dependent effects.

Gene KO	Pathway	Spectrum RT-Differentiation	Counts RT-Differentiation
**SBS**			
*DCLRE1A*	FA and ICL repair	LRR	LRR—LESS
*MLH1*	MMR	LRR	NA ^1^
*MSH2*	MMR	LRR	NA ^1^
*MSH6*	MMR	LRR	NA ^1^
*PMS1*	MMR	LRR	NA
*PMS2*	MMR	LRR	NA ^1^
*POLE3*	DNA replication	LRR	NA ^2^
*PRKDC*	NHEJ and MMEJ	LRR	NA
*REV1*	TLS	LRR	LRR—LESS
*TP53*	Checkpoint/DSB repair	LRR	NA
**INDELs**			
*PARP1*	BER/DSB repair/NER	-	ERR *—MORE
*PMS1*	MMR	-	LRR—MORE
*POLB*	BER	-	ERR—MORE
*POLE4*	DNA replication	-	LRR—LESS

For each gene, the primary DNA repair pathway is indicated, along with its impact on mutational spectrum and burden for both SBS (upper section) and INDELs (lower section) in early- and late-replicating timing regions. NA: No significant change (corrected *p*-value > 0.05) in either RT region. NA ^1^: No significant difference between RT states since there is a significant increase in mutation burden in both RT regions. NA ^2^: No significant difference between RT states since there is a significant decrease in mutation burden across both RT regions. *: Value near the significance threshold (*p* < 0.07). For comprehensive results, see Appendix A.

**Table 2 ijms-26-07307-t002:** Compounds exhibiting replication timing-dependent effects.

Mutagen	Group	Spectrum RT-Differentiation	Counts RT-Differentiation
**SBS**			
Formaldehyde (120 μM)	Aldehydes	NA	LRR *—MORE
MMS (100 μM)	Alkylating Agents	NA	ERR *—MORE
4-ABP (300 μM) + S9	Aromatic Amines	NA	ERR *—MORE
AZD7762 (1.625 μM)	DNA Damage Response Inhibitors	NA	ERR *—MORE
Olaparib (0.625 μM)	Drug Therapy	NA	LRR—LESS
N-Nitrosopyrrolidine (50 mM)	Nitrosamine	NA	LRR—MORE
DBA (75 μM) + S9	PAHs	LRR *	NA ^1^
DBADE (0.0313 μM)	PAHs	ERR *	NA ^1^
DBP (0.0039 μM)	PAHs	NA	LRR *—MORE
SSR (1.25 J)	Radiation	LRR *	NA ^1^
AAII (3.75 μM)	Others	NA	ERR—MORE
Furan (100 mM) + S9	Others	NA	ERR *—MORE
Methyleugenol (1.25 mM)	Others	NA	LRR—MORE
Styrene oxide (75 μM)	Others	NA	ERR—MORE
**INDELs**			
Cisplatin (12.5 μM)	Drug Therapy	-	ERR *—MORE
BPDE (0.125 μM)	PAHs	-	ERR *—MORE
BaP (2 μM) + S9	PAHs	-	ERR *—MORE
DBADE (0.0313 μM)	PAHs	-	ERR *—MORE
DBADE (0.109 μM)	PAHs	-	ERR *—MORE
DBP (0.0313 μM) + S9	PAHs	-	ERR—MORE

For each compound (grouped by mechanism of action), the table summarizes its impact on mutational spectrum and burden for both SBS (upper section) and INDELs (lower section) across early and late replication timing regions. NA: No significant change (adjusted *p*-value > 0.05) in any RT region. NA ^1^: No significant difference between RT states since there is a significant increase in mutation burden across both RT regions. *: Significant change in either ERR or LRR, with a marginal change in the other region. For comprehensive results, see Appendix A.

## Data Availability

The original contributions presented in this study are included in the article/Appendix A. Further inquiries can be directed to the corresponding authors.

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
