# Peer review of "Unveiling Replication Timing-Dependent Mutational Biases: Mechanistic Insights from Gene Knockouts and Genotoxins Exposures"

_ijms, 2025, doi:10.3390/ijms26157307_

Round 1

Reviewer 1 Report

Comments and Suggestions for Authors

The authors presents a novel approach to understanding how DNA replication timing (RT) influences mutagenesis across the genome. By integrating whole-genome sequencing data from CRISPR-Cas9 gene knockouts and mutagen exposures in human induced pluripotent stem cells (hiPSCs), the authors uncover significant replication timing-dependent mutational biases. The identification of 14 genes and 19 compounds with distinct RT-specific mutagenic effects provides valuable mechanistic insights into genome instability, particularly relevant for cancer development.

Major

While the study clearly demonstrates that RT is a key modulator of mutagenesis, the mechanistic explanation remains somewhat speculative. For example, the authors mention several DNA repair pathways but do not fully explore how these pathways are specifically modulated by replication timing in different genomic regions. More mechanistic experiments, such as protein expression or localization studies across the cell cycle, would help substantiate these claims. Validation of key RT-dependent mutagenic effects in more complex in vivo models would be beneficial to confirm the observed findings in human-relevant contexts.

Thearticle  integrates genetic knockouts with mutagen exposures to dissect mutational biases. However, the authors could expand on how these findings compare with real-world cancer mutations. For instance, do the mutational biases observed in these controlled experiments correlate with the mutational signatures found in human cancers? A comparative analysis would strengthen the clinical relevance of the findings.

The analysis of environmental mutagens is interesting, but the RT-dependent effects of these agents are discussed somewhat superficially. The manuscript could benefit from a deeper discussion of why certain compounds preferentially affect early-replicating regions (ERR) or late-replicating regions (LRR). Do these findings suggest a mechanistic explanation related to chromatin accessibility, repair efficiency, or DNA damage persistence?

While the statistical methods used are generally sound, the manuscript could benefit from a more detailed explanation of the rationale behind the bootstrapping approach. Furthermore, although the computational analysis is robust, some of the findings (for the POLQ knockout, for example) would benefit from experimental validation using more complex in vivo models or assays that better reflect the RT-specific mutagenic effects observed.

While the authors acknowledge limited sample sizes, a power analysis or justification for the sample size used would provide greater confidence in the robustness of the findings. Given the significance of RT-dependent effects, larger or additional datasets would help substantiate the observed trends.

Discuss how the RT-dependent mutational signatures from this study correlate with those observed in human cancers could provide deeper insights into the clinical implications of replication timing and the use of specific inhibitors or RNA interference targeting key DNA repair pathways to understanding of how these pathways are modulated by RT in both early- and late-replicating regions.

minor

The figures are clear but would benefit from higher resolution and more detailed legends, particularly in Fig. 2, where the panels describing RT-specific mutational burdens and spectra could be made more readable.

Please make sure that you use abbreviations (ERR, LRR, SBS) and define them clearly at first mention, particularly in the abstract and methods sections.

I consider the paper is generally well-written, but several sections (introduction and methods) could benefit from simplification for greater clarity. Some sentences are overly complex and could be broken into smaller, more digestible parts.

Author Response

We would like to thank you and the reviewers for the thorough evaluation of our manuscript titled:
"Unveiling Replication Timing-Dependent Mutational Biases: Mechanistic Insights from Gene Knockouts and Genotoxin Exposures"
We are encouraged by the reviewers’ overall positive assessment of our work and appreciate their thoughtful comments, which helped us improve the clarity and depth of the manuscript. Below, we respond point-by-point to each reviewer’s comments. All changes have been incorporated in the revised manuscript, and relevant sections are referenced accordingly.

Reviewer #1
Comment 1: While the study clearly demonstrates that RT is a key modulator of mutagenesis, the mechanistic explanation remains somewhat speculative. For example, the authors mention several DNA repair pathways but do not fully explore how these pathways are specifically modulated by replication timing in different genomic regions. More mechanistic experiments, such as protein expression or localization studies across the cell cycle, would help substantiate these claims. Validation of key RT-dependent mutagenic effects in more complex in vivo models would be beneficial to confirm the observed findings in human-relevant contexts.
Response: We appreciate the reviewer’s insightful suggestion to include mechanistic experiments such as protein expression or localization studies across the cell cycle. Our analyses indeed highlight strong candidate factors involved in RT-dependent mutagenesis, and we fully agree that understanding their dynamic regulation during S phase would provide further mechanistic insight. However, these experiments require significant time and resources and fall beyond the scope of this bioinformatics-focused study. We plan to pursue such experiments in future work. A paragraph outlining these future directions has been added to the Discussion section.

Comment 2: The article integrates genetic knockouts with mutagen exposures to dissect mutational biases. However, the authors could expand on how these findings compare with real-world cancer mutations. For instance, do the mutational biases observed in these controlled experiments correlate with the mutational signatures found in human cancers? A comparative analysis would strengthen the clinical relevance of the findings.
Response: We appreciate this important suggestion. We have explored this direction using several publicly available cancer genome datasets. However, we were unable to detect clear and interpretable patterns comparable to those observed in our controlled knockout models. The intrinsic heterogeneity of tumor genomes—harboring numerous mutations and overlapping mutational processes—makes it extremely difficult to isolate the contribution of individual genes. This challenge reinforces the value of our controlled, isogenic systems for dissecting replication timing–dependent mutational mechanisms. These limitations and interpretations are now discussed more explicitly in the revised Discussion section.

Comment 3: The analysis of environmental mutagens is interesting, but the RT-dependent effects of these agents are discussed somewhat superficially. The manuscript could benefit from a deeper discussion of why certain compounds preferentially affect early-replicating regions (ERR) or late-replicating regions (LRR). Do these findings suggest a mechanistic explanation related to chromatin accessibility, repair efficiency, or DNA damage persistence?
Response: We thank the reviewer for highlighting the need for a more thorough discussion of this point. We have now expanded the relevant section of the Discussion to include potential mechanistic explanations, including the role of chromatin accessibility, DNA repair efficiency, and lesion persistence. We discuss how these factors may underlie the observed differences in RT-dependent mutagenesis across various compounds.

Comment 4: While the statistical methods used are generally sound, the manuscript could benefit from a more detailed explanation of the rationale behind the bootstrapping approach.
Response: We have included in the results section a better explanation of the rationale for using bootstrapping in our analysis.

Comment 5: Some of the findings (e.g., for the POLQ knockout) would benefit from experimental validation using more complex in vivo models or assays that better reflect the RT-specific mutagenic effects observed.
Response: Please see our response to Comments 1 and 2. We acknowledge this limitation and have clarified it in the revised Discussion, while outlining how future studies may address these points.

We decided to omit POLQ from the paper based on the power analysis discussed in the next comment.

Comment 6: While the authors acknowledge limited sample sizes, a power analysis or justification for the sample size used would provide greater confidence in the robustness of the findings.
Response: While the sample size is beyond our control (we are using existing data), the idea of performing power analysis is very good. We have now included such analysis, based on simulation, in the results section. This approach confirmed that both knockout lines and compounds showing significant changes in mutation burden (P ≤ 0.05) are indeed robust and reliable. However, some of the cases we initially classified as non-significant may reflect limited statistical power, as they reached significance in a substantial fraction (>50%) of the simulations (Supplementary Figure 1 and Supplementary Table). As a consequence of this analysis we decided that the results regarding POLQ are not convincing and therefore we omitted them from the paper.

Comment 7: While the authors acknowledge limited sample sizes, a power analysis or justification for the sample size used would provide greater confidence in the robustness of the findings. Given the significance of RT-dependent effects, larger or additional datasets would help substantiate the observed trends.
Response: In the current study, we focused on iPSCs due to their pluripotent nature, which makes them broadly representative of many normal tissue types. While additional datasets exist, each containing data about a few KOs in Hap1 cells (a haploid leukemic cell line; Zou et al., Nature Communications 2018) or RPE1 cells (hTERT-immortalized retinal pigment epithelial cells; Koh et al., Nature Genetics, 2025) they are more limited in scope. We have already begun analyzing these datasets and plan to compare the association with RT found in those cells to the association found in iPSCs in a forthcoming publication.

Comment 8: Discussing how the RT-dependent mutational signatures from this study correlate with those observed in human cancers could provide deeper insights into the clinical implications of replication timing and the use of specific inhibitors or RNA interference targeting key DNA repair pathways.
Response: As addressed in Comment 2, we attempted such comparisons but found limited interpretability due to tumor genome complexity. This limitation is now discussed more explicitly in the manuscript.

Minor Comments

  • The figures are clear but would benefit from higher resolution and more detailed legends, particularly in Fig. 2.
    Response: Done.
  • Please define abbreviations (ERR, LRR, SBS) clearly at first mention, particularly in the abstract and methods.
    Response: Done.
  • Some sections (Introduction and Methods) could benefit from simplification for clarity.
    Response: Done.

Reviewer 2 Report

Comments and Suggestions for Authors

The authors of the manuscript titled “Unveiling Replication Timing-Dependent Mutational Biases: Mechanistic Insights from Gene Knockouts and Genotoxin Exposures” showed the correlation between genomic regions of different replication timing and genes involved in DNA replication/repair pathways as well as environmental mutagens. The analysis pipeline is clearly stated, and the results are highly informative for future research about detailed mutagenetic pathways. The only minor comment is that I hope the authors can publicize the data set in the future. It would be very valuable for other people to locate the genomic sequences that are caused by replication/repair gene knockouts or environmental mutagens.

Author Response

Comment: The only minor comment is that I hope the authors can publicize the data set in the future. It would be very valuable for other people to locate the genomic sequences that are caused by replication/repair gene knockouts or environmental mutagens.
Response: We thank the reviewer for this supportive comment. We did not generate new mutation data for this study but relied on published datasets, all of which are publicly available. Our analyses based on these data are provided in the supplementary materials for transparency and reuse.

Round 2

Reviewer 1 Report

Comments and Suggestions for Authors

The authors have answered the questions raised and in my opinion have improved their manuscript, omitting the part of POLQ that was not convincing.